# Olfactory Sensilla and Olfactory Genes in the Parasitoid Wasp *Trichogramma pretiosum* Riley (Hymenoptera: Trichogrammatidae)

**DOI:** 10.3390/insects12110998

**Published:** 2021-11-05

**Authors:** Basman H. Al-Jalely, Wei Xu

**Affiliations:** 1Food Futures Institute, Murdoch University, Murdoch, WA 6150, Australia; bassman_jalely@yahoo.com; 2College of Agricultural Engineering Sciences, University of Baghdad, Baghdad 10071, Iraq

**Keywords:** insect olfaction, expression profile, *Trichogramma pretiosum*, odorant binding protein, odorant receptor

## Abstract

**Simple Summary:**

Parasitic wasps are the major natural enemies of many organisms, and therefore they are broadly used in the biological control of numerous agricultural and horticultural pests. For example, *Trichogramma pretiosum* Riley (Hymenoptera: Trichogrammatidae) is a tiny natural egg parasitoid of various agricultural pest insects, including *Plutella xylostella*, *Helicoverpa armigera*, *Spodoptera frugiperda* and *Ectomyelois ceratoniae*. However, how *T. pretiosum* seek and localise host insect eggs is still not clear. The olfactory system is critical in guiding insect behaviours, including mating, feeding and oviposition, which play pivotal roles in the interactions between parasitoid wasps and their hosts. This project aimed to investigate *T. pretiosum* major olfactory tissue (antennae) and the olfactory genes, including odorant binding proteins (OBPs) and odorant receptors (ORs). *T. pretiosum* adult antennae were examined under scanning electron microscopy, and four types of olfactory sensilla were observed. Using *T. pretiosum* genome, 22 OBPs and 105 ORs were identified, which were further compared with olfactory genes of other Hymenoptera insect species. The expression patterns of OBPs between *T. pretiosum* male and female adults were examined to identify female- or male-specific OBPs. This study enriches our knowledge of *T. pretiosum* olfactory system and will help better use it in the integrated pest management (IPM) for many insect pest species.

**Abstract:**

*Trichogramma pretiosum* Riley (Hymenoptera: Trichogrammatidae) is a tiny natural egg parasitoid of several agricultural pest insects, which has been widely used in the biological control for *Plutella xylostella*, *Helicoverpa armigera*, *Spodoptera frugiperda* and *Ectomyelois ceratoniae*. However, limited studies have been conducted on *T. pretiosum* olfactory system, which is critical in regulating insect behaviours. In this study, *T. pretiosum* adult antennae were investigated under ascanning electron microscopy (SEM). Four types of olfactory sensilla were observed, including chaetica sensilla (CS), trichoid sensilla (TS), faleate sensilla (FS) and placoid sensilla (PS). Using *T. pretiosum* genome, 22 putative odorant binding proteins (OBPs) and 105 odorant receptors (ORs) were identified, which were further compared with olfactory genes of *Apis mellifera*, *Nasonia vitripennis* and *Diachasma alloeum*. The expression patterns of OBPs between *T. pretiosum* male and female adults were examined by quantitative real time PCR (qRT-PCR) approaches. Three female-specific OBPs (TpreOBP19, TpreOBP15 and TpreOBP3) were identified, which may play crucial roles in *T. pretiosum* host-seeking and oviposition behaviours. This study enriches our knowledge of *T. pretiosum* olfactory genes and improves our understanding of its olfactory system.

## 1. Introduction

Insect behaviours, including mating, foraging, host-finding and oviposition, are guided by their olfactory systems [1]. Hair-like olfactory sensilla distributed over the surface of antennae are utilised by insects to detect chemical signals from the environment [2,3,4]. With the advance of molecular and cellular biology, our understanding of insect olfactory mechanism has progressed. Various gene families have been reported to play pivotal roles in the dynamics, selectivity and sensitivity of insect olfactory systems, including odorant binding proteins (OBPs) [5], odorant receptors (ORs) [6,7], ionotropic receptors (IRs) [8], sensory neuron membrane proteins (SNMPs) [9] and odorant-degrading enzymes (ODEs) [5,10]. Hydrophobic odorants lack solubility, so they have difficulties in passing through the aqueous sensillum lymph and reaching receptors. OBPs, one class of proteins that are highly expressed in insect antennae, can bind, solubilise and deliver semiochemical molecules to ORs [11,12]. ORs are localised on the dendritic membrane in insect olfactory sensilla, detecting odorant compounds and transducing olfactory signals to insect brains to regulate behaviours [13]. ODEs can degrade the odorant compounds after they activate ORs, which clean the olfactory system for the new cycle of detection [5,10,14]. OBPs and ORs are involved in the first step of odorant detection, which are the target proteins in this study.

Parasitic wasps develop in or on various life stages of other arthropod hosts, and chemical signals are critical in guiding their mating, host-seeking and oviposition behaviours. For example, after mating, female wasps utilise host-associated chemical cues, including host pheromones or herbivore-induced plant volatiles to localise hosts [15,16]. *Trichogramma pretiosum* is a minute (≤0.5 mm long) wasp and female adults lay eggs into a number of lepidopteran eggs, including *Plutella xylostella*, *Helicoverpa armigera*, *Spodoptera frugiperda* and *Ectomyelois ceratoniae*. After *T. pretiosum* eggs hatch, the larvae devour the developing caterpillar, pupate and grow into adult wasps inside the host eggs. Adult wasps emerge by chewing holes in the host eggs and are then ready to parasitise other moth eggs. *T. pretiosum* have been used to control caterpillar pests in a wide range of horticultural and field crops [17].

Generally, egg parasitoid wasps rely on chemical cues originating from the adult host, host products, or the plant that host feeding on to seek host eggs [18], which are a very inconspicuous host stage attacked by parasitic wasps. It was reported that *T. pretiosum* use volatile host pheromones to locate host eggs [19]. Volatiles of female *Helicoverpa zea* and synthetic *H. zea* female pheromone components increased parasitisation rates by *T. pretiosum* [19]. Later, follow-up studies demonstrated that *T. pretiosum* responds to calling *H. zea* females [20]. However, limited attention has been paid to *T. pretiosum* olfactory system, which plays a critical role in semiochemical detection and guiding *T. pretiosum* host-seeking behaviours. The completion of the whole genome sequences of *T. pretiosum* provides an invaluable resource for us to annotate and analyse olfactory genes in this minute parasitoid wasp [21].

In this study, firstly, *T. pretiosum* antennae and sensilla were investigated by using scanning electron microscopy (SEM). Then *T. pretiosum* genome database was used to annotate the key olfactory genes, including OBPs and ORs, which were further analysed using bioinformatics, phylogenetics and molecular approaches.

## 2. Materials and Methods

### 2.1. Insect Materials

*T. pretiosum* pupae were sourced from Bugs for Bugs™ (https://bugsforbugs.com.au/, accessed on 27 November 2018) and kept in the lab at 25 ± 1 °C, 70–80% relative humidity (R.H.) and 16:8 h (Light:Dark) photoperiod. Emerged adults were collected immediately using vacuum traps and anaesthetised using carbon dioxide (purity > 99.9%, moisture < 100 ppm) for five minutes, and then sexed under a stereomicroscope (Olympus Corporation, Tokyo, Japan) based on antennae structures. Emerged adults were allocated for RNA extraction and scanning electron microscopy (SEM) (Ted Pella Inc., Redding, CA, USA).

### 2.2. Total RNA Extraction

After sexing, collected *T. pretiosum* adults were immediately stored in liquid nitrogen and then homogenised using a disposable homogenising pestle (Sigma-Aldrich, St. Louis, MO, USA). *T. pretiosum* adult body sizes are tiny (<0.5 mm), so we did not dissect different tissues such as antennae and palps for RNA extraction. Total RNA was extracted using the Qiagen RNeasy mini kit (Qiagen, Valencia, CA, USA) following the manufacturer’s protocol. The purified total RNA was treated by DNase I (New England Biolabs, Ipswich, MA, USA) to remove genomic DNA, quantified, quality checked using NanoDrop™ ND-2000 (Thermo Scientific, Waltham, MA, USA) and stored at −80 °C in the Western Australia State Agricultural and Biotechnology Centre (SABC, Murdoch, Australia).

### 2.3. Scanning Electron Microscopy (SEM)

*T. pretiosum* adults were preserved in 3% glutaraldehyde in 0.025 M pH 7.0 phosphate buffer for 24 h and then used for fixation with a Pelco Biowave processor (Ted Pella Inc., Redding, CA, USA). A critical point drying apparatus (Polaron E3000, Quorum Technologies, Lewes, UK) was used in the preparation process at the critical point of CO_2_ = 31.1 °C and 1071 psi. Dried samples were mounted on SEM stubs under a SZH10 microscope (Olympus Corporation, Tokyo, Japan) and then sputtered with 10 nm gold using a Polaron Sputter coater SC 7640 (Quorum Technologies, Lewes, UK) with argon gas (pressure < 1 × 10^−^^2^ mbar, voltage = 1 kV). Samples were examined and photographed under a Zeiss 1555 VP-FESEM SEM instrument (ZEISS Australia, North Ryde, Australia) operated at 10 kV, high current, 10–12 mm working distance, and 30 µm aperture. Sample preparation and examination were conducted at the Centre for Microscopy, Characterization, and Analysis (CMCA) at the University of Western Australia. Five male and five female antennae were observed under SEM (n = 5).

### 2.4. Gene Identification and Phylogenetic Analysis

Genes encoding for *T. pretiosum* OBPs, and ORs in the genome (NCBI: PRJNA297592) were identified using BLAST (blastn) searches with reported *D. melanogaster* and *Apis mellifera* OBP and OR genes as queries, as previously described [22]. Extensive manual curation was then performed on the *T. pretiosum* genome according to exon/intron splice site of GT-AG rule. The identified OBP and OR amino acid sequences (Appendix A) were used for validation by NCBI blastp based on the identity and similarity to orthologous genes from other insects. All identified *T. pretiosum* OBP and OR amino acid sequences are available in an online Appendix A.

Encoded TpreOBPs (Appendix A) were aligned by MEGA-X, a bioinformatics software for sequence data analysis using default settings. Gap Opening Penalty (10.00) and Gap Extension Penalty (0.20) were used for multiple sequence ClustalW alignment with 30% Delay Divergent Cut-off. N-terminal signal peptides were predicted using SignalP 5.0 (http://www.cbs.dtu.dk/services/SignalP, accessed on 5 September 2021). Calculated molecular weights (MW) and isoelectric points (pIs) were obtained using the ExPASy proteomics server (http://www.expasy.org/tools/protparam.html, accessed on 5 September 2021) [23]. The amino acid sequences of TpreOBPs were used to search the best blast hit sequences from NCBI using blastp. The Exon/Intron graphics were generated with GSDS (http://gsds.cbi.pku.edu.cn/index.php, accessed on 5 September 2021).

TpreOBP amino acid sequences (Appendix A) were used to create an entry file for phylogenetic analysis in MEGA-X with *A. mellifera*, *Nasonia vitripennis* and *Diachasma alloeum* OBP protein sequences [24,25,26,27]. Firstly, the amino acid sequences were aligned using ClustalW alignment with default settings: Gap Opening Penalty (10.00), Gap Extension Penalty (0.20) and 30% Delay Divergent Cut-off. A Maximum Likelihood tree was then constructed using the default settings based on Jones-Taylor -Thornton (JTT) model and Nearest-Neighbor-Interchange (NNI) method. The same phylogenetic analysis approach was also used to investigate *T. pretiosum*, *A. mellifera*, *N. vitripennis* and *D. alloeum* ORs.

### 2.5. Quantitative RT-PCR (qRT-PCR)

*T. pretiosum* cDNA templates were prepared from total RNA samples using the SuperScript™ VILO™ cDNA Synthesis Kit (Invitrogen, Waltham, MA, USA) following the manufacturer’s protocol. Quantitative Real-Time PCR (qRT-PCR) was performed using gene-specific primers (Appendix A), which were designed using the Primer3web (version 4.1.0) software (http://bioinfo.ut.ee/primer3/, accessed on 10 August 2019). The reference housekeeping gene, glyceraldehyde-phosphate dehydrogenase (GAPDH), was selected here because it has been shown as one of the best candidate reference genes in qPCR analysis [28]. A 2-Step qPCR protocol was performed on Rotor-Gene Q-5 Plex (Qiagen, Valencia, CA, USA) using Power SYBR^®^ Green PCR Master Mix (Thermo Fisher Scientific, Waltham, MA, USA) as follows: 95 °C for 5 min, followed by 40 cycles of 95 °C for 10 s, 60 °C for 15 s, and 65 to 95 °C in increments of 1.0 °C for 5 s [29]. For each cDNA sample and primer set, reactions were run in triplicate, and average fluorescence Ct values were obtained. Relative expression levels were determined using the comparative 2^−ΔΔCt^ method for relative quantification [30]. Three biological replicates were performed. Statistical analysis was performed on the expression profiles between male and female adults using the Student’s *t*-test (SPSS version, IBM, Armonk, NY, USA).

## 3. Results

### 3.1. Antennae and Sensilla

*T. pretiosum* adult antennae are sexually dimorphic (Figure 1), as previously reported for *Trichogramma australicum* antennae [31]. Both male and female antennae consist of an elongated scape (Sc) with basal radicle (R), pedicel (P) and flagellum (F). The flagellum is differentiated into basal anelli (ring segments), funicle and apical club (C), which are all apparent in the female antenna (Figure 1a,c).

The female club segment is broadest at its midpoint, slightly tapered, curved and is apically blunt. The apex is flattened on the dorsal surface and slightly curved on the ventral surface. The club is covered with numerous types of antennal sensilla on the surfaces, where sensory neurons for the perception of smell, taste, sound, and touch are localised. The male antenna has a distinct thin first anellus. The second anellus and funicular segments are fused with the club to form an elongate tube-like structure (Figure 1b,d). The club is slightly curved with a blunt apex. The surfaces of the male’s scape and pedicel are like that of the female antenna, but the surface of the club is more irregularly corrugated and covered with numerous relatively long sensilla. Based on the shape, three types of sensilla were recognised from male adult antennae, and they are chaetica sensilla (CS), trichoid sensilla (TS) and placoid sensilla (PS) (Figure 1e,f). However, in female adult antennae, four types of sensilla were observed. Besides CS, TS and PS, a new type of sensilla, faleate sensilla (FS), was observed as well (Figure 1e), which was reported as female-specific sensilla in *T. australicum* [31]. The different antennae structures between male and female adults were used here to distinguish male and female adults.

### 3.2. Identification of TpreOBPs and TpreORs

A total of 22 OBPs and 105 ORs were identified from the *T. pretiosum* genome (Table 1), which were used to compare with the olfactory proteins from *A. mellifera, N. vitripennis* and *D. alloeum*, three other Hymenoptera insects with available genome sequences. The number of OBPs (22) in *T. pretiosum* is higher than *D. alloeum* (15 OBPs), *A. mellifera* (21 OBPs), but lower than *N. vitripennis* (90 OBPs). The number of ORs (105 ORs) in *T. pretiosum* is significantly lower than in *A. mellifera* (163 ORs), *D. alloeum* (187 ORs), and *N. vitripennis* (301 ORs).

### 3.3. Bioinformatics and Phylogenetic Analysis of TpreOBPs

All 22 TpreOBPs are full-length sequences and exhibit 66–99% identity to known insect OBPs at the amino acid level. For example, TpreOBP2 showed 99% identify to OBP2 (Sequence ID ANG08492.1) from *Trichogramma*
*dendrolimi*. TpreOBP3 showed 97% identify to OBP3 (Sequence ID ASA40277.1) from *Trichogramma japonicum*. No signal peptide was predicted from TpreOBP7, while all the other 21 TpreOBPs carry signal peptides. The mature (without signal peptide) TpreOBPs range from 109 to 132 amino acids, and their molecular weights range from 12,279 to 15,033 Da (Table 2). The isoelectric points (pIs) of most TpreOBPs are below 7.0 except TpreOBP1, TpreOBP14, TpreOBP18 and TpreOBP20, whose pIs are higher than 7.0 (Table 2). The alignment of TpreOBP amino acid sequences highlights the six conserved cysteine residues (Figure 2a). Most of *T. pretiosum* OBPs share the characteristic features of the classic OBP family: small size, presence of an N-terminal signal peptide sequence as well as a highly conserved pattern of six different exon/intron structures were identified from 22 TpreOBP genes, which consist of two exons, four exons, five exons or six exons, respectively (Figure 2b). TpreOBP18 is the only OBP gene consisting of two exons. TpreOBP11 and TpreOBP17 contain four exons in each. TpreOBP2, 4, 6, 7, 8, 13, 14, 16, 20, 21, and 22, contain five exons while TpreOBP1, 3, 5, 9, 10, 12, 15 and 19 contain six exons.

The phylogenetic analysis of OBPs was performed among *T. pretiosum* and three other Hymenopteran species: *A. mellifera, N. vitripennis* and *D. alloeum* (Figure 3). On the phylogenetic tree, various TpreOBPs were clustered closely with *N. vitripennis* OBPs (NvitOBPs) (Figure 3), suggesting they share high identities at the amino acid level. For example, TpreOBP11 and NvitOBP26 share 68.4% identity. TpreOBP18 and NvitOBP18 share 63.4% identity. These TpreOBPs may play similar roles as clustered NvitOBPs. For example, host-seeking, oviposition or detecting plant compounds for nectar feeding. TpreOBP7 belongs to “plus-C” because it contains seven cysteines in the mature sequence. All other *T. pretiosum* OBPs belong to the “classic” OBPs, while no “Minus-C” or “Double” OBPs were annotated from *T. pretiosum* (Figure 3). There are two “Minus-C” OBP groups: one was formed by AmelOBP13-21 while the other was formed by NvitOBP56-62 and NivtOBP27 (Figure 3). One “double” OBP group was formed by NvitOBP38-46, and NvitOBP48 was observed from the tree (Figure 3), as described before.

### 3.4. Phylogenetic Analysis of T. pretiosum ORs

The phylogenetic analysis of 105 TpreORs, 163 AmelORs, 301 NvitORs and 187 DallORs was performed in a similar way as OBPs. The results revealed a number of species-specific OR subfamilies (Figure 4), which may point to species-specific adaptations during evolution and lifestyles. For example, a *T. pretiosum*-specific OBP was observed, which did not show similarity to any other ORs from *A. mellifera, N. vitripennis* and *D. alloeum*. As one of the most conserved genes in various insect species, TpreORCO shared 72.8% identity with *N. vitripennis* ORCO (NvitOR1), 60.8% identity with *A. mellifera* ORCO (AmelOR2) and 51.3% identity with *D. alloeum* ORCO (DallOR1) at the amino acid level. Various TpreORs were clustered closely with *N. vitripennis* ORs (NvitORs) (Figure 4), just as observed in TpreOBPs. TpreOR63 and NvitOR105 share 50.6% identity.

### 3.5. Expression Profiles of T. pretiosum OBPs

OBPs are involved in the first step of odorant detection, so they are our major targets in the expression profile study. Expression profiles between male and female insects can help identify the female- or male-specific OBPs and build the links between OBPs and their potential functions. Here, TpreOBPs were examined between male and female adults by qRT-PCR. To test the designed primers, reverse transcriptase (RT)-PCR was conducted first, and the products were analysed using electrophoresis. All 22 TpreOBPs were successfully amplified, and their band sizes were the same as expected, suggesting the primers were designed appropriately.

To study the expression levels, the qRT-PCR approach was utilised to compare TpreOBP expression levels between males and females with the TpreGADPH gene as a reference gene (Figure 5). The results were presented in three types: Type 1 are female-specific OBPs, which showed the expression levels of OBPs in female adults are over five times higher than in male adults (female/male ratio > 5), including TpreOBP19, TpreOBP15 and TpreOBP3 (Figure 5a). For example, the female/male (F/M) ratio of the expression of TpreOBP19 was 42.6 ± 13.8. The F/M ratio of the expression of TpreOBP15 was 23.7 ± 2.9, while the F/M ratio of the expression level of TpreOBP3 was 23.1 ± 1.4. 

Type 2 are male-specific OBPs, including TpreOBP22, TpreOBP5, TpreOBP10 and TpreOBP17, which demonstrated significantly higher expression levels (>5) in male adults than in female adults (Figure 5b). For example, the male/female (M/F) ratio of the expression of TpreOBP22 is 167.7 ± 61.9. The F/M ratio of the expression level of TpreOBP5 was 66.0 ± 3.2. The M/F ratio of the expression level of TpreOBP10 was 18.4 ± 0.8, while the M/F ratio of the expression level of TpreOBP17 was 8.7 ± 0.8. All other OBPs belong to type 3 because either the F/M ratio or M/F ratio of the expression level is lower than 5 (F/M < 5 or M/F < 5) (Figure 5c).

## 4. Discussion

Parasitoid wasps constitute a large group of hymenopteran superfamilies, which lay eggs on or in the bodies of other arthropods, resulting in the death of their hosts. Trichogrammatids are some of the smallest parasitoid wasps that grow up within-host insect eggs. One of them, *T. pretiosum*, is frequently used as biological control agent against major lepidopteran pests, including *P. xylostella*, *H. armigera* and *E. ceratoniae*. However, the mechanism of how *T. pretiosum* localise host eggs is not fully understood yet. Without this knowledge, it is difficult for us to better understand *T. pretiosum* host-seeking and parasitising behaviours and refine our current application by using it as an efficient and effective biological control agent. This study on *T. pretiosum* olfactory system provided us with insights into this field.

Firstly, *T. pretiosum* adult antennae were examined using SEM because they are the major insect olfactory organs. Male and female *T. pretiosum* showed very different antennae structures. Furthermore, the faleate sensilla (FS) was only observed in female but not male antennae, suggesting significant differences between their olfactory systems. Interestingly, this was not observed in another *P. xylostella* parasitoid wasp, *Diadegma semiclausum*, which lays eggs into *P. xylostella* larvae specifically (Unpublished data).

Completing the *T. pretiosum* genome project is a significant step towards further understanding its olfactory system and potential applications in pest control. *T. pretiosum* genome shows rapid genome evolution compared to other hymenopterans (Pereira et al., 2019), reflecting adaptations to miniaturisation and to life as a specialised egg parasitoid. In the *T. pretiosum* genome, 22 OBPs and 105 ORs were identified, and compared with olfactory proteins of *A. mellifera*, *N. vitripennis* and *D. alloeum*. *A. mellifera* is one of the model species for Hymenoptera, whose olfaction and social behaviours have been extensively studied. *N. vitripennis* is the most widely studied of the parasitoid wasps, which is a generalist and parasitises a wide range of dipteran hosts, including blowflies, fleshflies and houseflies. *D. alloeum* is a specialist parasitoid of the fruit fly, *Rhagoletis pomonella*. The number of OBPs and ORs exhibited significant differences in these four Hymenopteran species, supporting the differences among their chemosensory systems and biology. For example, *A. mellifera* has 163 ORs but only ten gustatory receptors (GRs). Presumably a large number of ORs can enhance *A. mellifera* olfactory ability and facilitate the typical foraging and social behaviours while honeybees have limited need for GRs for plant secondary metabolite discrimination since flowering plants have evolved visual and olfactory cues to attract bees [26]. *T. pretiosum* has the lowest numbers of ORs in these four species. It may be due to that most of its life stages are inside the host eggs and its tiny body size. A number of OBPs and ORs exhibited high similarities at amino acid level between *T. pretiosum* and *N. vitripennis*, suggesting these proteins may play similar roles in both two generalist parasitoid wasps. A recent study compared the OBPs and ORs between *T. pretiosum* and *D. semiclausum* because both are important parasitoid wasps and widely used in the biological control for *P. xylostella*, one of the most destructive insect pests of cruciferous plants [32]. Unlike *T. pretiosum*, a natural egg parasitoid that lay eggs to various host insect eggs, *D. semiclausum* lay eggs into *P. xylostella* larvae specifically and selectively. A number of OBPs and ORs were identified showing high similarities between these two wasps (e.g., TpreOR39 and DsemOR30, TpreOR33 and DsemOR19, TpreOR77 and DsemOR63, TpreOBP2 and DsemOBP8), which may function similarly in these two species. More conserved OBPs or ORs will be identified from *D. semiclausum* after its genome was available. Further functional characterisation of these OBPs or ORs will provide insights to their roles in *T. pretiosum* olfactory system and behaviours.

OBPs are involved in the first step of odorant detection, so the characterisation of their sex-specific expression will help build the links between OBPs and their potential functions. For example, female-specific OBPs are more likely to play a significant role in the parasitism behaviours. Male-specific OBPs may be involved in the sex pheromone detection and mating behaviours. Here the whole bodies of male and female *T. pretiosum* adults were used for RNA extraction rather than antennae because *T. pretiosum* is too tiny (<0.5 mm), and the antennae collection is extremely time-consuming, which may cause RNA degradation. Therefore, the qRT-PCR results only exhibited the expression profile of TpreOBPs between male and female adult bodies, but not antennae. OBPs have been reported to be expressed in non-olfactory organs rather than antennae, and they may have other functions [12]. For example, it was reported that *Aedes aegypti* OBP22 is produced in the sperm and transferred to females during mating [33]. Therefore, further localisation of TpreOBPs will be needed by using technologies such as in situ hybridisation or immunohistochemistry.

TpreOBP19, 15 and 3, are female-specific OBPs, which are the candidate OBPs to assist in parasite behaviours. TpreOBP22, TpreOBP5, TpreOBP10 and TpreOBP17 showed male-specific expression, suggesting they may contribute to female seeking and mating behaviours. Other TrepOBPs showed similar expression levels between male and female *T. pretiosum*, indicating they may play similar roles in both. For example, they are detecting flower odors for nectar-sucking, which are a major sugar resource for insect species. These results help us link the functions of *T. pretiosum* OBPs for future functional characterisation studies. Further, in vitro (e.g., ligand binding assay) and in vivo (e.g., RNAi or CRISPR) functional characterisation will help demonstrate their roles in *T. pretiosum* olfactory system. For example, after the knock-down or knock-out of the target OBPs or ORs, the treated insect responses and behaviours will be observed to various chemical compounds [34,35]. On the other hand, the candidate OBPs can be used by a “reverse chemical ecology” approach to identify their ligands and demonstratetheir functions [36].

## 5. Conclusions

In summary, 22 OBPs and 105 ORs were identified from the genome sequence of *T. pretiosum*, which were further studied by phylogenetic and bioinformatics methods. The sex-specific expression patterns of *T. pretiosum* OBPs were analysed using qRT-PCR between male and female adults. This study advances our understanding of the chemosensory system of *T. pretiosum* at the molecular level and provides a foundation for further research on the olfactory system in *T. pretiosum*.

## Figures and Tables

**Figure 1 insects-12-00998-f001:**
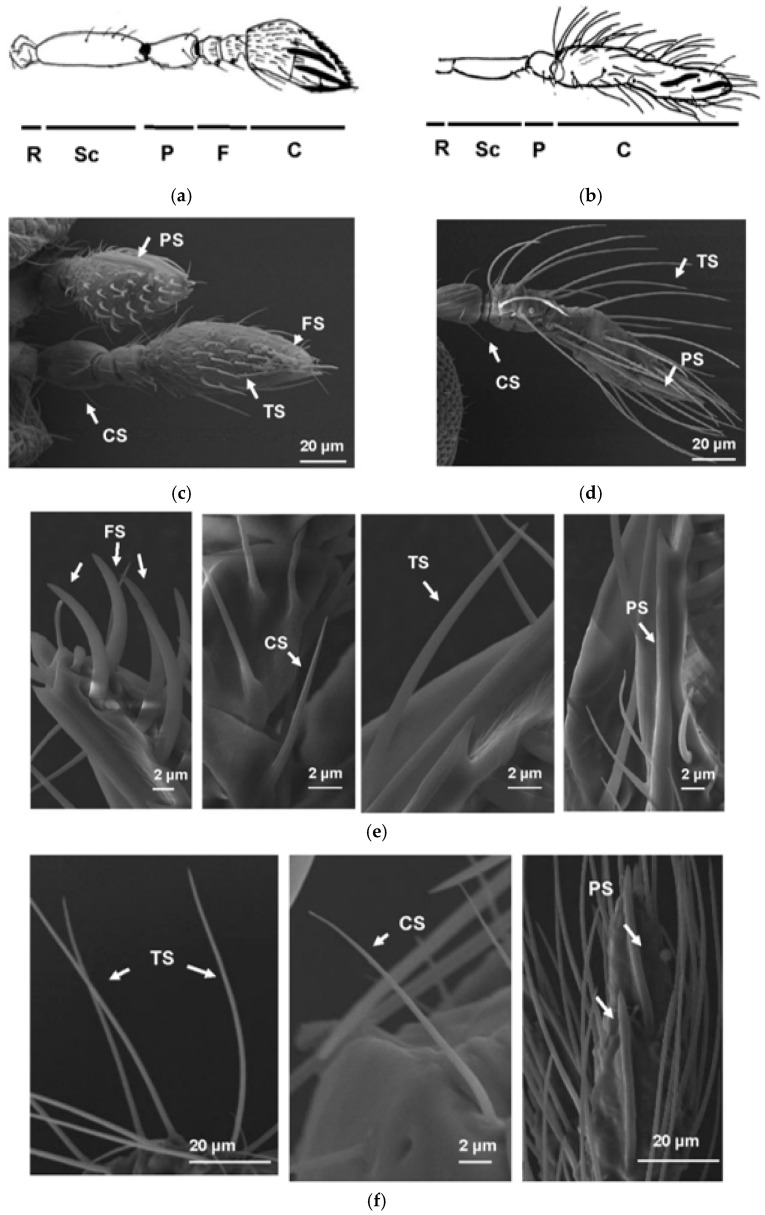
The scanning electron microscopy (SEM) analysis of *T. pretiuosum* female and male antennae and sensilla. (**a**,**b**) the hand drawings of *T. pretiuosum* female and male antennae. (**c**,**d**), SEM analysis of *T. pretiuosum* female and male antennae. (**e**,**f**), SEM analysis of *T. pretiuosum* female and male olfactory sensilla on the antennae. Sc, elongated scape; R, basal radicle; P, pedicel; F, flagellum; and C, apical club. Various types of olfactory sensilla were observed, including chaetica sensilla (CS), trichoid sensilla (TS), Placoid sensilla (PS) and faleate sensilla (FS).

**Figure 2 insects-12-00998-f002:**
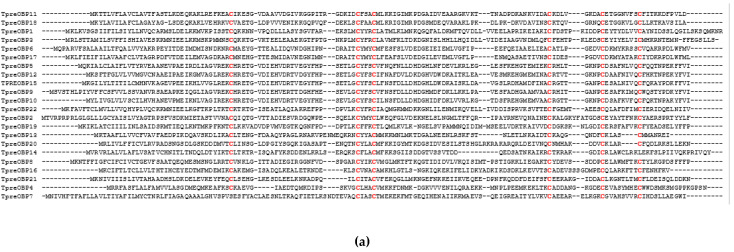
The alignment of 22 TpreOBP amino acid sequences (**a**) and their intron/exon gene structures (**b**). The six conserved cysteines are highlighted by red colour. Black blocks represent exons, while the lines represent introns.

**Figure 3 insects-12-00998-f003:**
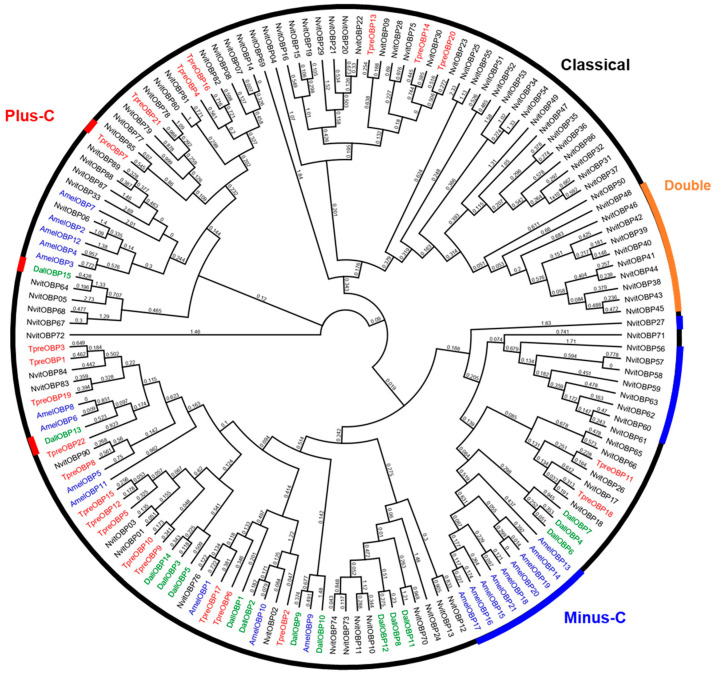
Phylogenetic analysis of OBPs from *T. pretiosum* (TpreOBPs), *A. mellifera* (AmelOBPs), *N. vitripennis* (NvitOBPs) and *D. alloeum* (DallOBPs). 22 TpreOBPs are marked in red, 21 AmelOBPs are marked in blue, 90 NvitOBPs are marked in black, and 15 DallOBPs are marked in green. Consensus support values are labelled on branches.

**Figure 4 insects-12-00998-f004:**
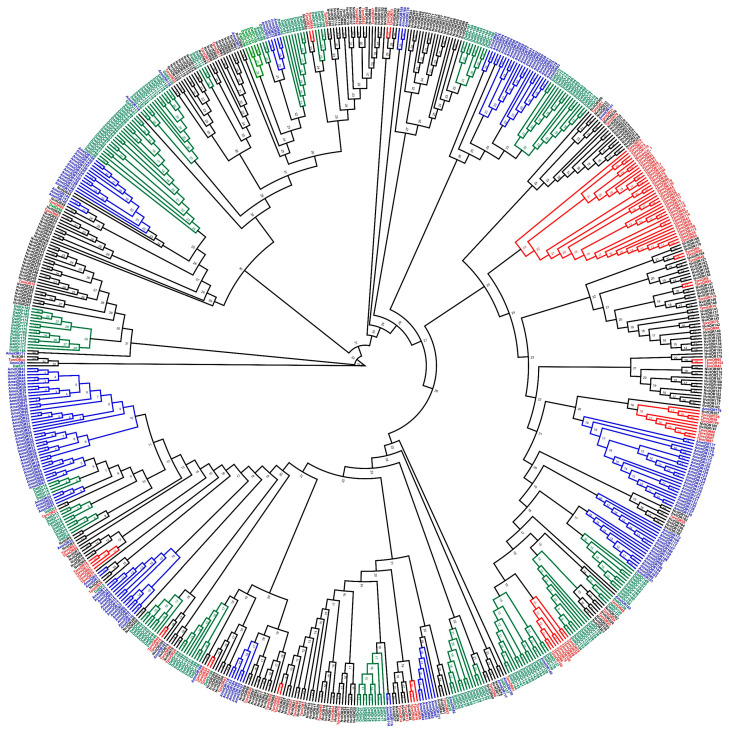
Phylogenetic analysis of ORs from *T. pretiosum* (TpreORs), *A. mellifera* (AmelORs), *N. vitripennis* (NvitORs) and *D. alloeum* (DallORs). 105 TpreORs are marked in red, 163 AmelORs are marked in blue, 301 NvitORs are marked in black, and 187 DallORs are marked in green. The ORCO (*) subfamily formed by TpreORCO, NvitOR1, DallOR1 and AmelOR2 are highlighted.

**Figure 5 insects-12-00998-f005:**
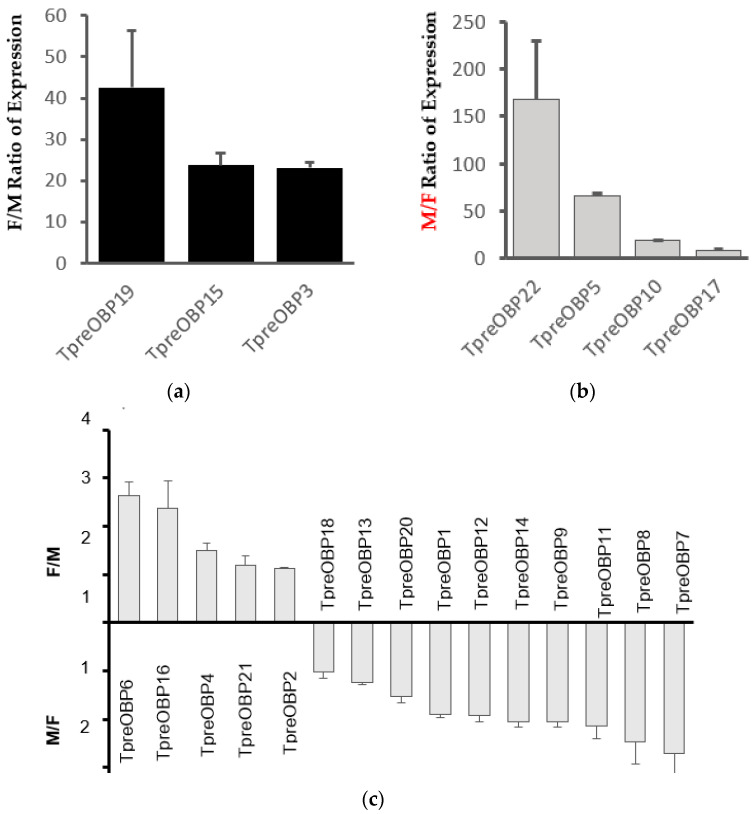
Quantitative real-time PCR (qPCR) analysis of *T. pretiosum* OBPs (TpreOBPs) from female and male adults. Normalised by TpreGADPH gene. (**a**), the female/male (F/M) expression ratios of TpreOBP19, TpreOBP5 and TpreOBP3, which are female-specific OBPs with F/M > 5. (**b**), the male/female expression ratios of TpreOBP22, TpreOBP5, TpreOBP10 and TpreOBP17, which are male-specific OBPs with M/F > 5. (**c**), other TpreOBPs that are expressed in both male and female adults with F/M or M/F ratios were less than 5.0. Error bars show standard deviation. M, adult males and F, adult females.

**Table 1 insects-12-00998-t001:** The numbers of OBPs and ORs, in *Trichogramma pretiosum*, *Apis mellifera, Nasonia vitripennis*, and *Diachasma alloeum*.

Species	OBP	OR
*Trichogramma pretiosum*	22	105
*Apis mellifera*	21	163
*Nasonia vitripennis*	90	301
*Diachasma alloeum*	15	187

**Table 2 insects-12-00998-t002:** *T. pretiosum* OBPs.

Gene Name	Full Length	Signal Peptide	Isoelectric Points PI	Molecular Weight	Mature Amino Acids	Expect Value	Ident	Sequence ID
TpreOBP1	Yes (Y)	Y, 1–22	8.71	15,032.73	132	2E-91	90%	ANG08491.1 odorant-binding protein 1 [*T. dendrolimi*]
TpreOBP2	Y	Y, 1–25	4.83	14,599.41	127	2E-107	99%	ANG08492.1 odorant-binding protein 2 [*T. dendrolimi*]
TpreOBP3	Y	Y, 1–22	5.96	14,983.56	130	5E-106	97%	ASA40277.1 OBP3 [*T. japonicum*]
TpreOBP4	Y	Y, 1–19	5.45	12,776.75	114	8E-84	90%	ANG08494.1 odorant-binding protein 4 [*T. dendrolimi*]
TpreOBP5	Y	Y, 1–19	5.14	13,830.53	120	2E-92	93%	ANG08495.1 odorant-binding protein 5 [*T. dendrolimi*]
TpreOBP6	Y	Y, 1–23	4.15	13,825.72	120	6E-99	96%	ANG08496.1 odorant-binding protein 6 [*T. dendrolimi*]
TpreOBP7	Y	No	5.56	14,358.55	131	2E-44	63%	CCD17854.1, putative odorant binding protein 85 [*Nasonia vitripennis*]
TpreOBP8	Y	Y, 1–22	6.46	13,407.55	122	3E-98	97%	ANG08498.1, odorant-binding protein 8 [*T. dendrolimi*]
TpreOBP9	Y	Y, 1–27	5.36	13,361.11	119	2E-103	99%	ANG08499.1, odorant-binding protein 9 [*T. dendrolimi*]
TpreOBP10	Y	Y, 1–18	5.63	13,547.40	119	4E-90	93%	ANG08500.1, odorant-binding protein 10 [*T. dendrolimi*]
TpreOBP11	Y	Y, 1–17	7.48	12,666.67	119	9E-88	98%	ANG08501.1, odorant-binding protein 11 [*T. dendrolimi*]
TpreOBP12	Y	Y, 1–18	5.27	13,426.28	119	2E-96	98%	ANG08502.1, odorant-binding protein 12 [*T. dendrolimi*]
TpreOBP13	Y	Y, 1–17	7.8	12,279.11	109	1E-53	66%	XP_014219837.1, uncharacterized protein LOC106647812 [*Copidosoma floridanum*]
TpreOBP14	Y	Y, 1–19	9.32	13,439.53	117	1E-92	97%	ANG08504.1, odorant-binding protein 14 [*T. dendrolimi*]
TpreOBP15	Y	Y, 1–19	5.06	13,402.08	119	2E-97	97%	ANG08505.1, odorant-binding protein 15 [*T. dendrolimi*]
TpreOBP16	Y	Y, 1–19	4.96	12,914.79	114	4E-82	92%	ANG08506.1, odorant-binding protein 16 [*T. dendrolimi*]
TpreOBP17	Y	Y, 1–20	4.04	13,483.34	120	4E-91	93%	ANG08507.1, odorant-binding protein 17 [*T. dendrolimi*]
TpreOBP18	Y	Y, 1–17	8.95	12,678.93	117	2E-89	0.98	AZB49386.1, odorant-binding protein 5 [*Heortia vitessoides*]
TpreOBP19	Y	Y, 1–20	5.5	13,980.46	123	2E-81	0.98	ANG08509.1, odorant-binding protein 19 [*T. dendrolimi*]
TpreOBP20	Y	Y, 1–16	7.75	13,005.76	118	6E-68	0.99	ANG08510.1, odorant-binding protein 20, partial [*T. dendrolimi*]
TpreOBP21	Y	Y, 1–16	4.34	13,959.62	121	2E-88	0.96	ANG08512.1, odorant-binding protein 22 [*T. dendrolimi*]
TpreOBP22	Y	Y, 1–23	6.53	14,056.71	123	1E-82	0.82	ASA40280.1, OBP6 [*T. japonicum*]

## Data Availability

The datasets during and/or analysed during the current study are available from the corresponding author on reasonable request.

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
