# Peer review of "Olfactory Sensilla and Olfactory Genes in the Parasitoid Wasp Trichogramma pretiosum Riley (Hymenoptera: Trichogrammatidae)"

_insects, 2021, doi:10.3390/insects12110998_

Round 1
Reviewer 1 Report
In this manuscript, the authors characterize the olfactory sensilla of the parasitoid wasp T. pretiosum and identify the OBPs and ORs present in the genome of T. pretiosum. The study was completed by comparing the expression of OBP in male and female using quantitative PCR.
The manuscript adds new informations concerning the chemosensory system of T. pretiosum.
However, the manuscript has many errors and problems requiring extensive proofreading. Here is the non-exhaustive list of errors and problems noted:
line 40: reference is not numbered
line 42: wrong reference numbering: the references 1, 2 3 are missing
line 50: The sentence :" The first insect OBP was...." is not very useful
line 56: odorant detection
Result part: Species and genus insect names mentioned in this part of the manuscript must be written in italic.
Fig. 2: Wrong emplacements of (A) & (B)
Fig. 4: Unreadable figure: improve the quality of the figure in order to be able to zoom it
line 303: Fig. 5 instead of Fig. 7
In Fig. 5 (B): M/F ratio instead of F/M ratio
line 337: DBM signification ?
line 341: numbered the reference
line 344 & 350: wrong reference numbering
Author Response
In this manuscript, the authors characterize the olfactory sensilla of the parasitoid wasp T. pretiosum and identify the OBPs and ORs present in the genome of T. pretiosum. The study was completed by comparing the expression of OBP in male and female using quantitative PCR.The manuscript adds new informations concerning the chemosensory system of T. pretiosum.
However, the manuscript has many errors and problems requiring extensive proofreading. Here is the non-exhaustive list of errors and problems noted:
line 40: reference is not numbered!
Corrected! Now all references are numbered.
line 42: wrong reference numbering: the references 1, 2 3 are missing
Corrected! See above.
line 50: The sentence :" The first insect OBP was...." is not very useful
Deleted!
line 56: odorant detection
Corrected!
Result part: Species and genus insect names mentioned in this part of the manuscript must be written in italic.
Done!
Fig. 2: Wrong emplacements of (A) & (B)
Corrected!
Fig. 4: Unreadable figure: improve the quality of the figure in order to be able to zoom it
Done! A new high quality figure was used to replace the old one.
line 303: Fig. 5 instead of Fig. 7
Done!
In Fig. 5 (B): M/F ratio instead of F/M ratio
Done!
line 337: DBM signification ?
Replaced by using P. xylostella.
line 341: numbered the reference
Done! See above!
line 344 & 350: wrong reference numbering
Done! Corrected!
Reviewer 2 Report
Review ID 1433424
Identification and characterization of olfactory genes in the parasitoid wasp Trichogramma pretiosum Riley (Hymenoptera: Trichogrammatidae)
This is very well organized manuscript. I recommend accepting it for printing in its current form.
Gene research is always important. It brings us closer to discovering their functions. It rarely happens, but it happens that the article does not raise any controversy. This is what this manuscript is like.
The conclusions are consistent with the evidence and arguments presented and they address the main question posed.
The references are appropriate.
I like Tables, Figures and photos. I have no objections to them.
Author Response
Thanks to all you comments.
Reviewer 3 Report
The manuscript of Al-Jalely and Xu aims at providing some more information on the olfactory genes of Trichogramma pretiosum. Even though the topic itself is rather interesting, I believe that the authors have not really managed to deliver a manuscript tyhat can be accepted for publication in its current form. Even though from the title itself, someone would expect a purely bioinformatic paper, the authors initiate the manuscript with some very nice (I have to admit) but relatively unexpected pictures of antennae and sensilla of T. pretiosum adults. Though one might argue that these pictures are relevant as the manuscript refers to the olfactory system of T. pretiosum, the authors have stated with their title that they would study the olfactory system at the genome-level and not with regard to morphological traits. As a consequence, the reader is somehow distracted from the main scope of the manuscript. Regarding the identification of olfactory genes, the authors align the published genome of T. pretiosum with other hymenopteran species, and then proceed with a direct comparison with them. However, this alone does not suffice for a publication as the authors should significantly enhance the discussion and present some data that would make the publication worth publishing. For that reason, I regret to inform you that this manuscript should be rejected in its current form. The authors however, are encouraged to re-structure the manuscript in a manner that would be more consistent with the initial scope of their study, and of course, enhance it in that direction. Finally there are also some linguistic errors as well that should also be corrected (see the list below - this is not an exhaustive list, but it simply contains some points that I spotted when going through the text).
Below you can find some points that the authors should pay particular attention to, when re-structuring the manuscript for a future re-submission.
- Lines 9-34: Simple Summary and Abstract are very similar - please try to change the wording.
- Line 39: ...are manipulated...
- Lines 69-72: rephrase this sentence as the use of the term "host" is used for two different organisms (host plant and host insect) and it becomes difficult to understand the meaning each time.
- Line 75: what do you mean here with the "calling"?
- Line 280 and forth: every Latin name should be in italics and that should be corrected throughout the text.
- Line 303: The sentence that "Only female adult T. pretiosum..." lay eggs, sounds very weird as this is something common for most insects (personally speaking I am not aware of many species whose males lay eggs).
- Line 309: ...(F/M) ratio...
- Line 343:...with olfactory proteins of...
- Line 345: Did you mean here "supporting" instead of "suggesting"?
- Lines 327-363: The discussion needs a thorough enhancement, as in its current context it is rather simple, presenting only some superficial differences in the number of olfactory genes between T. pretiosum and other species. To really support such a publication, the authors should elaborate more on the use of each gene and really decipher their involvement in the olfactory system of T. pretiosum.
Author Response
Below you can find some points that the authors should pay particular attention to, when re-structuring the manuscript for a future re-submission.
- Lines 9-34: Simple Summary and Abstract are very similar - please try to change the wording.
The simple summary was re-written
2. Line 39: ...are manipulated...
Done!
3. Lines 69-72: rephrase this sentence as the use of the term "host" is used for two different organisms (host plant and host insect) and it becomes difficult to understand the meaning each time.
This sentence was rewritten!
4. Line 75: what do you mean here with the "calling"?
The release of pheromone by female moths is known as calling and is characterized by the female raising her abdomen to better disperse pheromone from the scent gland.
5. Line 280 and forth: every Latin name should be in italics and that should be corrected throughout the text.
Done!
6. Line 303: The sentence that "Only female adult T. pretiosum..." lay eggs, sounds very weird as this is something common for most insects (personally speaking I am not aware of many species whose males lay eggs).
Deleted!
7. Line 309: ...(F/M) ratio...
Done!
8. Line 343:...with olfactory proteins of...
Done!
9. Line 345: Did you mean here "supporting" instead of "suggesting"?
Done!
10. Lines 327-363: The discussion needs a thorough enhancement, as in its current context it is rather simple, presenting only some superficial differences in the number of olfactory genes between T. pretiosum and other species. To really support such a publication, the authors should elaborate more on the use of each gene and really decipher their involvement in the olfactory system of T. pretiosum.
The discussion part was rewritten according to reviewers’ comments.
Round 2
Reviewer 3 Report
Even though I still feel that the manuscript containts two different parts that are not appropriately connected with one another (sensilla and genes), the authors have succesfully tackled with the points and issues raised. Given that there are not many things that could be done, I believe that it can now be accepted for publication.